# Three-dimensional simulation of the pancreatic parenchyma, pancreatic duct and vascular arrangement in pancreatic surgery using a deep learning algorithm

**Ryoichi Miyamoto**[1]*, **Amane Takahashi**[1], **Aya Ogasawara**[2], **Toshiro Ogura**[1], **Kei Kitamura**[1], **Hiroyuki Ishida**[1], **Shinichi Matsudaira**[1], **Satoshi Nozu**[3], **Yoshiyuki Kawashima**[1]

1 Department of Gastroenterological Surgery, Saitama Cancer Center, Kita-Adachi-gun, Saitama, Japan, 2 Imaging Technology Center, Fujifilm Corporation, Minato-ku, Tokyo, Japan, 3 Department of Radiology, Saitama Cancer Center, Kita-Adachi-gun, Saitama, Japan

* goodfirst883@gmail.com

**Data Availability Statement:** All relevant data are within the paper and its Supporting information files.

## Abstract

Three-dimensional surgical simulation, already in use for hepatic surgery, can be used in pancreatic surgery. However, some problems still need to be overcome to achieve more precise pancreatic surgical simulation. The present study evaluates the performance of SYNAPSE VINCENT® (version 6.6, Fujifilm Medical Co., Ltd., Tokyo, Japan) in the semiautomated surgical simulation of the pancreatic parenchyma, pancreatic ducts, and peripancreatic vessels using an artificial intelligence (AI) engine designed with deep learning algorithms. One-hundred pancreatic cancer patients and a control group of 100 nonpancreatic cancer patients were enrolled. The evaluation methods for visualizing the extraction were compared using the Dice coefficient (DC). In the pancreatic cancer patients, tumor size, position, and stagewise correlations with the pancreatic parenchymal DC were analyzed. The relationship between the pancreatic duct diameter and the DC, and between the manually and AI-measured diameters of the pancreatic duct were analyzed. In the pancreatic cancer/control groups, the pancreatic parenchymal DC and pancreatic duct extraction were 0.83/0.86 and 0.84/0.77. The DC of the arteries (portal veins/veins) and associated sensitivity and specificity were 0.89/0.88 (0.89/0.88), 0.85/0.83 (0.85/0.82), and 0.82/0.81 (0.84/0.81), respectively. No correlations were observed between pancreatic parenchymal DC and tumor size, position, or stage. No correlation was observed between the pancreatic duct diameter and the DC. A positive correlation (r = 0.61, p<0.001) was observed between the manually and AI-measured diameters of the pancreatic duct. Extraction of the pancreatic parenchyma, pancreatic duct, and surrounding vessels with the SYNAPSE VINCENT® AI engine assumed to be useful as surgical simulation.

**Funding:** This research is a cooperative study with Fujifilm Medical Co., Ltd., Tokyo, Japan. MR received a specific grant from Fujifilm Medical Co., Ltd., Tokyo, Japan. The funders and MR analyzed the collected data together. The funder had no role in study design, data collection, decision to publish, or preparation of the manuscript.

**Competing interests:** I have read the journal's policy and the authors of this manuscript have the following competing interests: This research is a cooperative study with Fujifilm Medical Co., Ltd., Tokyo, Japan. This does not alter our adherence to PLOS ONE policies on sharing data and materials.

## Introduction

Precise recognition of anatomic variations and the relationships of tumors with the surrounding organs and vessels is required to safely perform hepatobiliary pancreatic surgery. Pancreatic surgeries, such as pancreaticoduodenectomy (PD), usually involve resection of the gastroduodenal artery and pancreaticoduodenal arcade, including the inferior pancreaticoduodenal artery, which can exhibit significant anatomic variation [1, 2]. Recognition of the hepatic arterial blood supply and anatomy of fragile veins, including the left gastric vein, inferior mesenteric vein and jejunal vein, are also essential to avoiding or minimizing the risk of surgical complications [3–5].

Although identifying the main pancreatic duct on the resected pancreatic surface is also crucial for dissecting pancreatic tissue or reconstructing the pancreatojejunostomy anastomosis, precise recognition of a nondilated main pancreatic duct is very difficult in normal (i.e., soft) pancreatic tissue because the pancreatic duct cannot be simulated by 2D multidetector computed tomography (MDCT) alone [6–8].

Recent advances in workstations for radiological diagnostic imaging have made it easier to construct 3D images of vasculature and organs. In 2013, we originally developed and expanded 3D surgical simulation technology that was already in use for hepatic surgery for pancreatic surgery [9, 10]. We previously reported that 3D surgical simulations in pancreatic surgery helped enhance the surgical staff's understanding of the surgical anatomy, improve perioperative factors, and predict pancreatic duct positions in the pancreatectomy planes and even the residual pancreatic volume [11–13].

However, some problems still need to be overcome to achieve more precise pancreatic surgical simulation. The surgical simulation analysis includes the highly variable and complex peripancreatic vessels, the effect of infiltration and inflammation of the cut surface on the pancreatic parenchyma and surrounding vessels, individual differences in the contrast-enhanced imaging of the pancreatic parenchyma, and accuracy of recognition when maneuvering in the pancreatic duct. For these reasons, high-precision automatic extraction using software has been difficult, and surgeons must manually modify these 3D images based on extensive experience and knowledge of peripancreatic anatomy.

Herein, the present study evaluates the performance of SYNAPSE VINCENT® (version 6.6, Fujifilm Medical Co., Ltd., Tokyo, Japan), which enables semiautomated surgical simulations of the pancreatic parenchyma, pancreatic ducts, and peripancreatic vessels using an artificial intelligence (AI) engine designed with deep learning algorithms.

## Materials and methods

### Patients

The AI engine for pancreatic analysis in SYNAPSE VINCENT® was designed using deep learning algorithms. In addition to the pancreatic parenchyma, pancreatic ducts, and peripancreatic vessels, the system can extract regions such as the spleen, kidney, and duodenum from dynamic MDCT images. We evaluated the extraction accuracy of this AI engine in 100 pancreatic cancer patients and a control group of 100 nonpancreatic cancer patients. The background characteristics of the 100 patients with pancreatic cancer are shown in Table 1. The evaluation methods for visualizing the pancreatic parenchyma and pancreatic duct were compared using the Dice coefficient (the ratio of the average number of elements in the two sets to the number of common elements; the closer the value is to 1, the higher the similarity). For vascular extraction (arteries, portal veins/veins), sensitivity and specificity were calculated along with the Dice coefficient, in addition to visual evaluation of the images by a senior pancreatic surgeon

**Table 1. The background characteristics of the 100 patients with pancreatic cancer.**

| Factor | |
|---|---|
| CT-Slice width (mm) | |
| 0.75 | 68 |
| 1.00 | 32 |
| Stage (UICC, 8th edition) | |
| 0 | 7 |
| IA | 18 |
| IB | 7 |
| IIA | 27 |
| IIB | 16 |
| III | 11 |
| IV | 14 |
| Tumor size (mm) | 26 (5.0–80) |
| Pancreatic duct size (mm) | 4 (1.0–15) |
| Location | 67 (60.5%) |
| Head | 56 |
| Body | 31 |
| Tail | 13 |

NLR, neutrophil-to-lymphocyte ratio; BMI, body mass index; ASA, American Society of Anesthesiologists; PS, performance status; CEA, carcinoembryonic antigen; CA19-9, carbohydrate antigen 19–9; CA72-4, carbohydrate antigen 72–4; AFP, α-fetoprotein; Well, well differentiated adenocarcinoma; Moderately, moderately differentiated adenocarcinoma; Poorly, poorly differentiated adenocarcinoma; UICC, Union for International Cancer Control; LADG, laparoscopy-assisted distal gastrectomy; B-I, Billroth-I; B-II, R-Y, Roux-en Y

*: $p < 0.05$

(three-stage evaluation, Fig 1). This study was approved by the ethics review committee of our institute and Fujifilm Medical Corporation. All data were fully anonymized before we accessed them; therefore, the ethics committee waived the requirement for informed consent in this retrospective study.

## 3D images used in the present study

Dynamic contrast-enhanced CT images in the arterial phase and portal vein phase with slick thicknesses from 0.75 to 1.00 mm were input into the SYNAPSE VINCENT® AI engine for pancreatic analysis. Arteries were extracted in the arterial phase, and the pancreas, pancreatic duct, and portal veins/veins were extracted in the portal vein phase (Fig 2A–2D). The correct mask was prepared using SYNAPSE VINCENT®, in which the masks for the pancreas, pancreatic duct, arteries, and portal veins/veins were manually marked by a senior pancreatic surgeon for important sites in the preoperative simulation of the pancreas.

## Evaluation of the accuracy of the extracted 3D images (Fig 3A and 3B)

To determine the similarity between the correct mask and extracted mask, the Dice similarity coefficient was calculated [14, 15]. This index evaluates the matching rate of elements in two sets and is defined below, where TP means true positives, FP means false positives, and FN

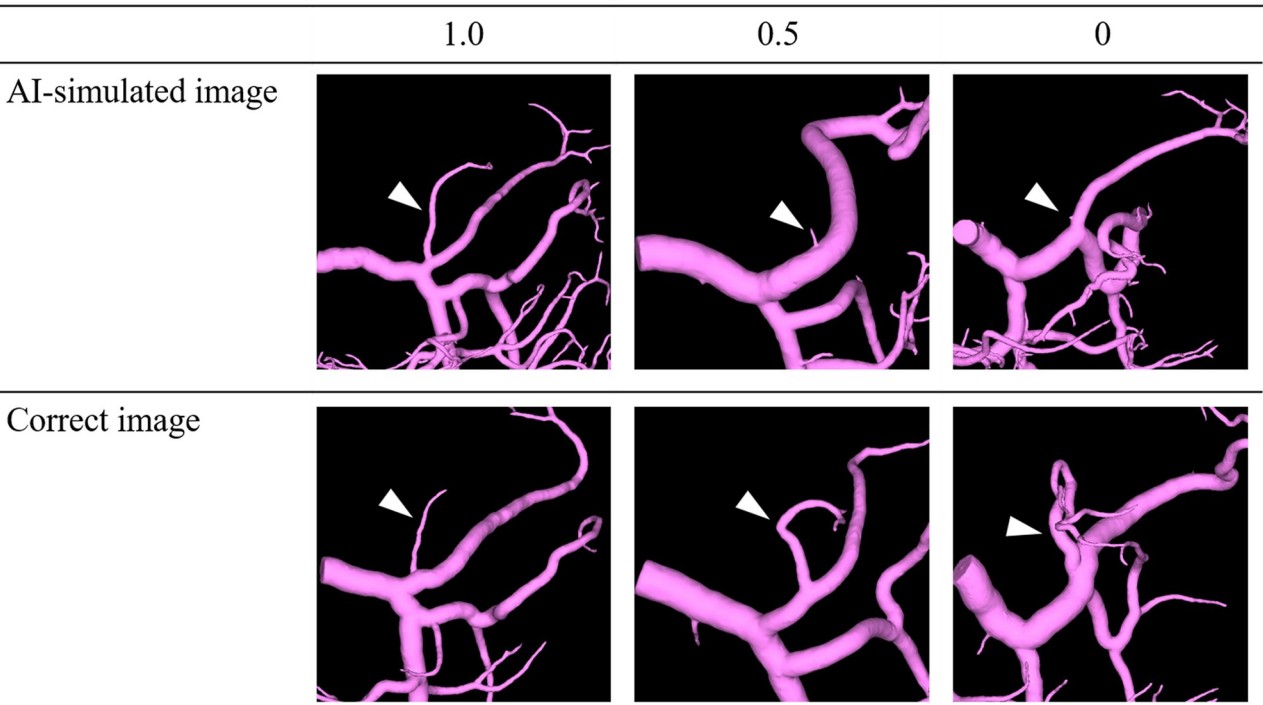

**Fig 1. Three-stage visual evaluation of the images by a senior pancreatic surgeon.** For arteries and portal veins/veins, sensitivity and specificity were calculated along with the Dice coefficient, in addition to visual evaluation of the images by a surgeon (three-stage evaluation). For visual evaluation, the sensitivity of the pancreatic surgeon in identifying each blood vessel was defined as 1.0 if the area required for the preoperative simulation could be extracted, 0.5 if there was any interruption or only the branch could be extracted, and 0.0 otherwise.

means false negatives:

$$DSC = \frac{2TP}{2TP + FP + FN}$$

It is important to correctly recognize the courses of the pancreatic ducts, arteries, and portal veins/veins. Therefore, the Dice coefficient was calculated based on the center voxel of the extracted mask. However, it is not possible to obtain a perfect match when comparing the center voxels of the correct mask and extracted mask, and there can be slight deviations that do not have a clinical impact. Therefore, $\delta$, which is equal to the average maximum radius of the correct mask, was introduced and defined as follows:

$$TP_c = G_c \cap (d(P_c) < \delta)$$

$$FP_c = (d(G_c) > \delta) \cap P_c$$

$$FN_c = G_c \cap (d(P_c) > \delta)$$

$$DSC_c = \frac{2TP_c}{2TP_c + FP_c + FN_c}$$

Here, $Gc$ is the center voxel set of the correct mask, $Pc$ is the center voxel set of the extracted mask, and $d(x)$ is the distance from set x. Although the AI engine can extract peripheral blood

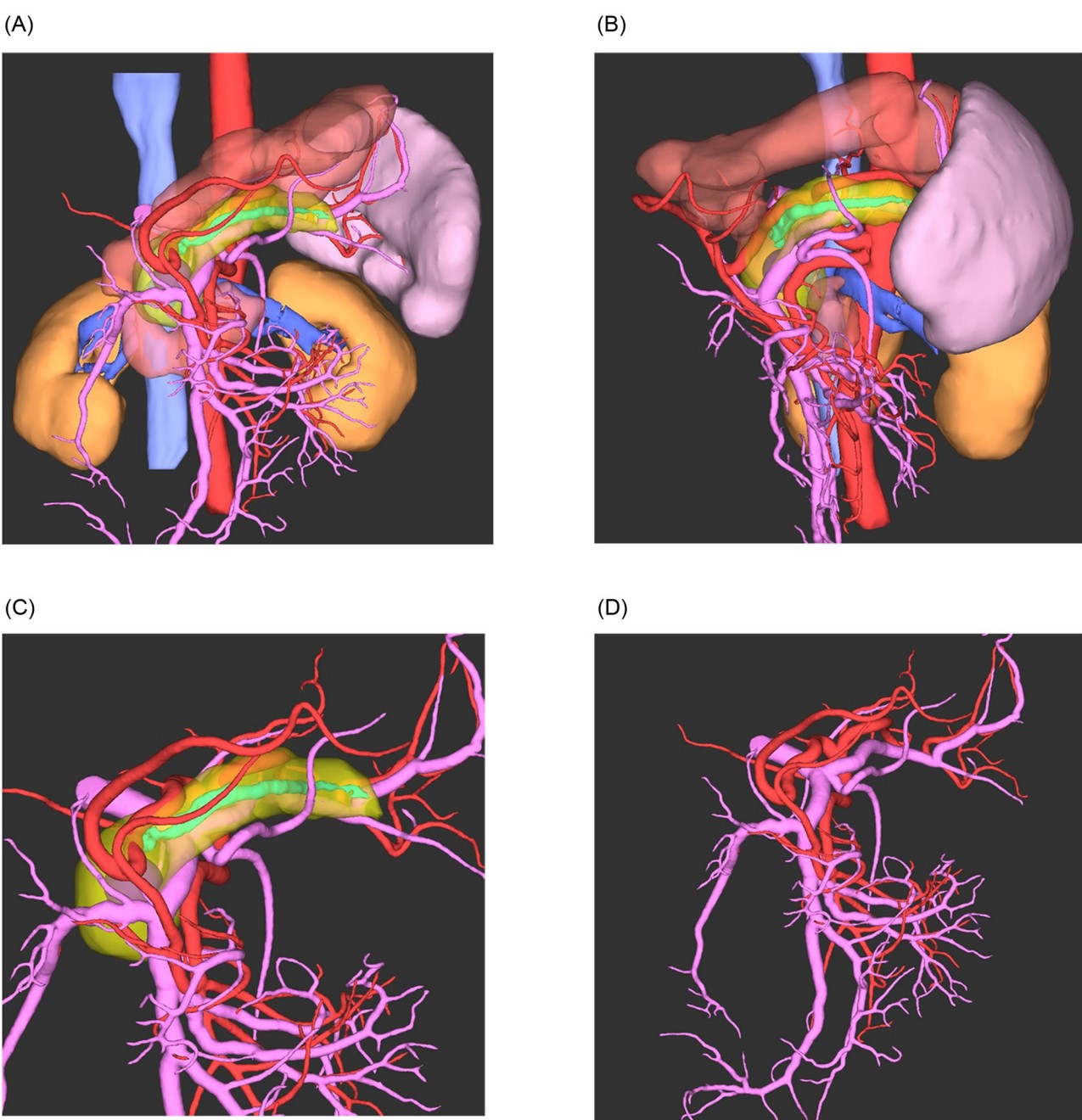

**Fig 2. A three-dimensional (3D) image from a pancreatic cancer patient.** The red color represents the arteries; pink represents the veins, including the portal vein; yellow represents the pancreas; turquoise represents the pancreatic duct; blue represents the renal vein and inferior vena cava; dark yellow represents the kidney; light pink represents the spleen; copper represents the stomach and duodenum; and dark pink represents the pancreatic tumor. A This view is an anterior 3D image. B This view is a left-sided lateral 3D image. C This view is an anterior 3D image only with pancreas, pancreatic duct, peripancreatic vessels including the artery and veins including the portal vein, and pancreatic tumor. D This view is an anterior 3D image only with the peripancreatic vessels, including the artery and veins including the portal vein.

vessels of low clinical importance in certain areas, the correct mask, extracted by the pancreatic surgeon, is applied only for important sites in the preoperative simulation. Therefore, *FPc* represents the peripheral blood vessels extracted from the analysis of the complete image, and the clinical value of the extracted arterial mask and portal vein/vein masks is underestimated.

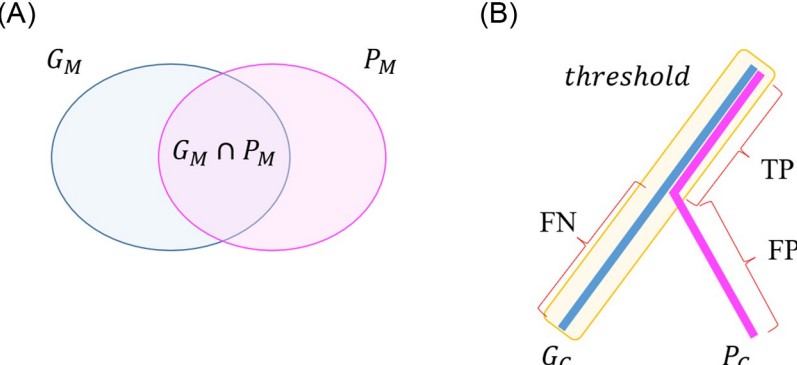

**Fig 3. Evaluation of the accuracy of the extracted 3D images using the Dice coefficient.** A The Dice similarity coefficient evaluates the matching rate of elements in two sets and is defined below, where TP means true positives, FP means false positives, and FN means false negatives. B The Dice similarity coefficient evaluates the matching rate of elements in two vessels data sets and is defined below, where Gc is the center voxel set of the correct mask, Pc is the center voxel set of the extracted mask, TP means true positives, FP means false positives, and FN means false negatives.

Thus, the vascular region outside the correct mask range, which does not have a clinical effect, was excluded from the calculation.

For arteries and portal veins/veins, sensitivity and specificity were calculated along with the Dice coefficient, in addition to visual evaluation of the images by a surgeon (three-stage evaluation). Sensitivity and specificity were defined as follows:

$$Sens = \frac{TP_c}{TP_c + FN_c}$$

$$Prec = \frac{TN_c}{TN_c + FN_c}$$

where *TNc* is the true-negative rate based on the center voxel; if the center voxel of the extracted arterial mask is not included in the δ of the correct portal vein/vein mask or the center voxel of the extracted portal vein/vein mask is not included in the δ of the correct arterial mask, δ is defined as *TNc*. With regard to the visual evaluation, the sensitivity of the pancreatic surgeon in identifying each blood vessel was defined as 1.0 if the area required for the preoperative simulation could be extracted, 0.5 if there was any interruption or only the branch could be extracted, and 0.0 otherwise. For the peripancreatic vessels to be visually evaluated, the arteries included the hepatic artery, splenic artery, gastroduodenal artery, left gastric artery, posterior inferior pancreatic duodenal artery, jejunal artery, and middle colic artery; the portal veins/veins included the jejunal veins, ileocolic veins, inferior mesenteric vein, gastrocolic venous trunk, right gastroepiploic vein, posterior superior pancreatic duodenal vein, accessory right colic vein, middle colic vein, and left gastric vein.

## Comparison of clinical data in pancreatic cancer patients

Among the patients with pancreatic cancer, the following were studied to examine the effectiveness of the image analysis, such as in identifying the pancreatic parenchyma of the tumor and vascular infiltration and the accuracy of AI extraction: tumor size, position (head, body, tail), and stagewise correlations between the Dice coefficient of the pancreatic parenchyma and the Dice coefficient of the main pancreatic duct, between the main pancreatic duct

diameter and the main pancreatic duct Dice coefficient, and between the manually measured diameter of the main pancreatic duct and the AI-measured diameter.

## Statistical analyses

The Dice coefficients for extraction of the pancreatic parenchyma, pancreatic duct, and vessels, including arteries, portal veins and veins, were compared using the unpaired t test and Fisher's exact test. Similarly, the sensitivity of visual inspection for peripancreatic vascular extraction and clinical data were compared using the unpaired *t* test and Fisher's exact test.

Correlations between tumor size and the Dice coefficient of the pancreatic parenchyma and between the main pancreatic duct diameter and main pancreatic duct Dice coefficient are presented as scatter plots and were analyzed using Pearson tests. Correlations between the diameters of the main pancreatic duct in 3D simulations and the corresponding actual diameters are presented as scatter plots and were analyzed using Pearson tests.

Statistical analyses were performed using a statistical analysis software package (SPSS Statistics, version 21; IBM, Armonk, NY, USA), and *p* values < 0.05 were considered significant.

## Result

### Accuracy of the extracted 3D images

Overall, the Dice coefficients for parenchymal and ductal extraction were 0.84 and 0.81, respectively. In the pancreatic cancer group/control group, the Dice coefficients of pancreatic parenchyma and pancreatic duct extraction were 0.83/0.86 (p = 0.001) and 0.84/0.77 (p = 0.001), respectively. Visualization of the pancreatic parenchyma was significantly better in the control group than in the pancreatic cancer group, and visualization of the pancreatic duct was significantly better in the pancreatic cancer group than in the control group (Table 2).

The overall Dice coefficients of the arteries and portal veins/veins were 0.89 and 0.88, respectively. In the pancreatic cancer group/control group, the Dice coefficient of the arteries (portal veins/veins) and associated sensitivity and specificity were 0.89/0.88 (p = 0.394), (0.89/0.88) (p = 0.275), 0.85/0.83 (0.85/0.82), and 0.82/0.81 (0.84/0.81), respectively. No obviously significant differences were found in the Dice coefficients between arteries and portal veins/veins.

In the pancreatic cancer group/control group, the sensitivity of visual inspection for peripancreatic vascular extraction is shown in Table 3, and the mean values were significantly different between the pancreatic cancer group and control group (0.86 and 0.97) (p<0.001); the values were particularly low in the posterior inferior pancreatic duodenal artery and posterior superior pancreatic duodenal vein in the pancreatic cancer group.

**Table 2. Accuracy of the extracted 3D images including pancreas, pancreatic duct, artery, vein.**

| | Dice coefficients | | | | Sensitivity | | | Specifity | |
|---|---|---|---|---|---|---|---|---|---|
| Factors | All | Pancreatic cancer | Control | *p value* | All | Pancreatic cancer | Control | Pancreatic cancer | Control |
| Pancreas | 0.84 | 0.83 | 0.86 | 0.001* | - | - | - | - | - |
| Pancreatic duct | 0.81 | 0.84 | 0.77 | 0.001* | - | - | - | - | - |
| Artery | 0.89 | 0.89 | 0.88 | 0.394 | 0.84 | 0.85 | 0.83 | 0.82 | 0.81 |
| Vein | 0.88 | 0.89 | 0.88 | 0.275 | 0.83 | 0.85 | 0.82 | 0.84 | 0.81 |

*: p<0.05

**Table 3. The sensitivity for peripancreatic vascular extraction in the two groups.**

| Vessels | Mean | Pancreatic cancer | Control | p value |
|---|---|---|---|---|
| Hepatic artery | 0.90 | 0.86 | 0.94 | 0.0265* |
| Splenic artery | 0.97 | 0.94 | 1.00 | 0.0011** |
| GDA | 0.97 | 0.94 | 1.00 | 0.0020** |
| LGA | 0.96 | 0.94 | 0.97 | 0.2786 |
| IPDA | 0.71 | 0.58 | 0.85 | 0.0000*** |
| Jejunal artery | 0.92 | 0.86 | 0.99 | 0.0001*** |
| MCA | 0.94 | 0.89 | 0.99 | 0.0003*** |
| Jejunal vein | 0.98 | 0.97 | 1.00 | 0.0179* |
| ICV | 0.99 | 0.98 | 1.00 | 0.0235* |
| IMV | 0.99 | 0.97 | 1.00 | 0.0127* |
| GCT | 0.99 | 0.97 | 1.00 | 0.0127* |
| RGEA | 0.97 | 0.94 | 1.00 | 0.0011** |
| ASPDV | 0.67 | 0.49 | 0.86 | 0.0000*** |
| Accessory RCV | 0.94 | 0.90 | 0.99 | 0.0003*** |
| MCV | 0.97 | 0.94 | 1.00 | 0.0005*** |
| LGV | 0.84 | 0.74 | 0.94 | 0.0001*** |

GDA, gastroduodenal artery; LGA, left gastric artery; IPDA, inferior pancreaticoduodenal artery; MCA, middle colic artery; ICV, iliocolic artery; inferior mesenteric vein; GCT, gastrocolic trunc; RGEA, right gastroepiproic artery; ASPDV, anterior superior pancreaticoduodenal vein; RCV, right colic vein; LGV. left gastric vein.

*:p<0.05,

**:p<0.01,

***:p<0.001

## Subgroup analysis of pancreatic cancer patients

No correlations were observed between the main pancreatic duct Dice coefficient or pancreatic parenchymal Dice coefficient and tumor size, position, or stage (Figs 4–6). No correlation was observed between the main pancreatic duct diameter and main pancreatic duct Dice coefficient (Fig 7). A positive correlation (r = 0.61, p<0.001) was observed between the manually measured main pancreatic duct diameter and the AI-extracted main pancreatic duct diameter, with an average error of -0.17 mm (Fig 8).

## Discussion

Based on the Dice coefficient, sensitivity, specificity, and visual evaluation results, the present study confirmed that the pancreatic parenchyma, pancreatic duct, and peripancreatic vessels can be extracted with high accuracy using SYNAPSE VINCENT®, which was designed using a deep learning algorithm. Furthermore, similar analyses on disease stage, tumor size, lesion site, and pancreatic duct diameter were conducted in the subgroup analysis of the pancreatic cancer group, but no deterioration in extraction accuracy was observed for any parameter.

Through visual confirmation of the 20 patients (pancreatic cancer group: 15 patients; control group: 5 patients) in the bottom 10% with significantly lower Dice coefficients of the pancreatic parenchyma, the cause of masking failure was found to be replacement of the pancreatic parenchyma by a tumor in eight cases (40%), atrophy of the pancreatic parenchyma in seven cases (35%), cystic lesions in four cases (20%), and other reasons in one case (5%). Because there was no difference in visualizing the pancreas in terms of position or size of the tumor, thinning of the pancreatic parenchyma due to cystic components and unclear

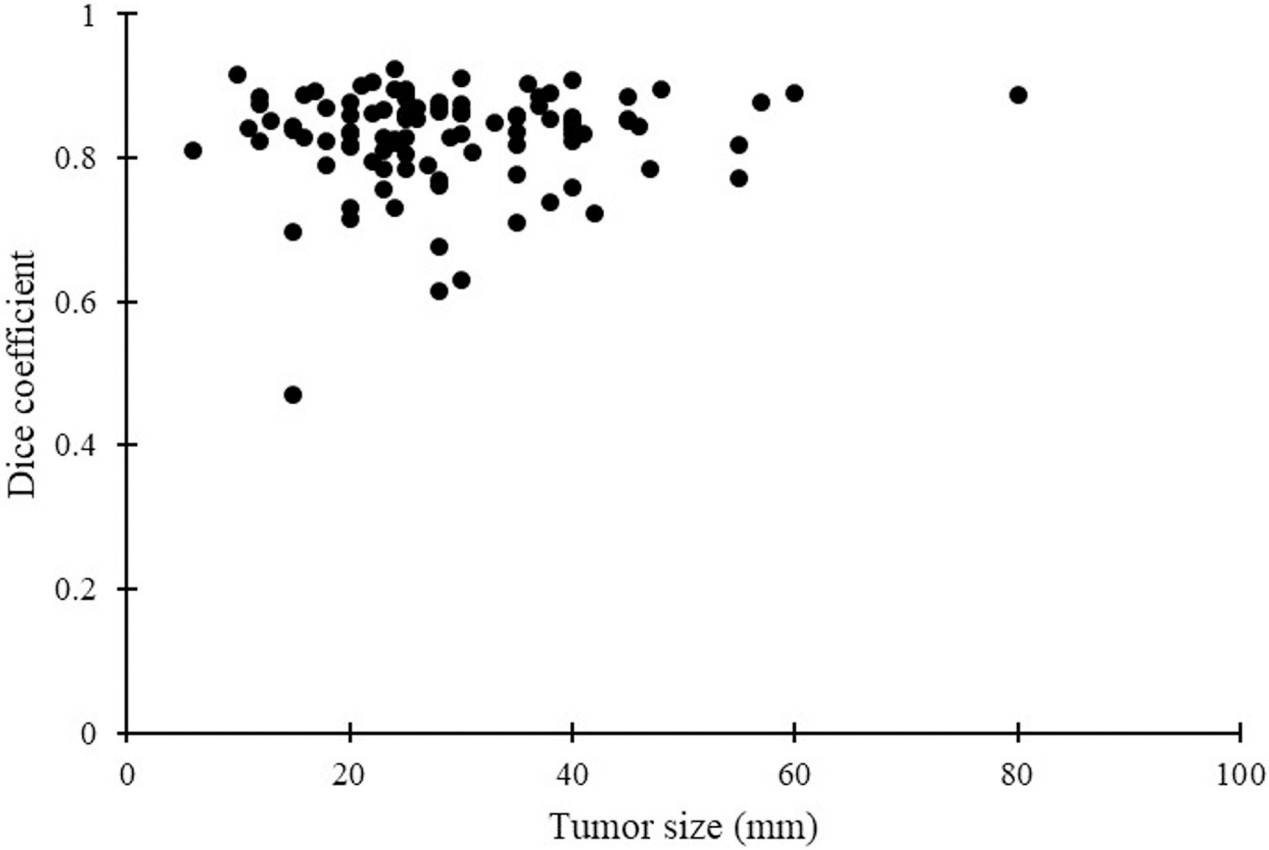

**Fig 4. Correlation between tumor size and Dice coefficient of the pancreas.**

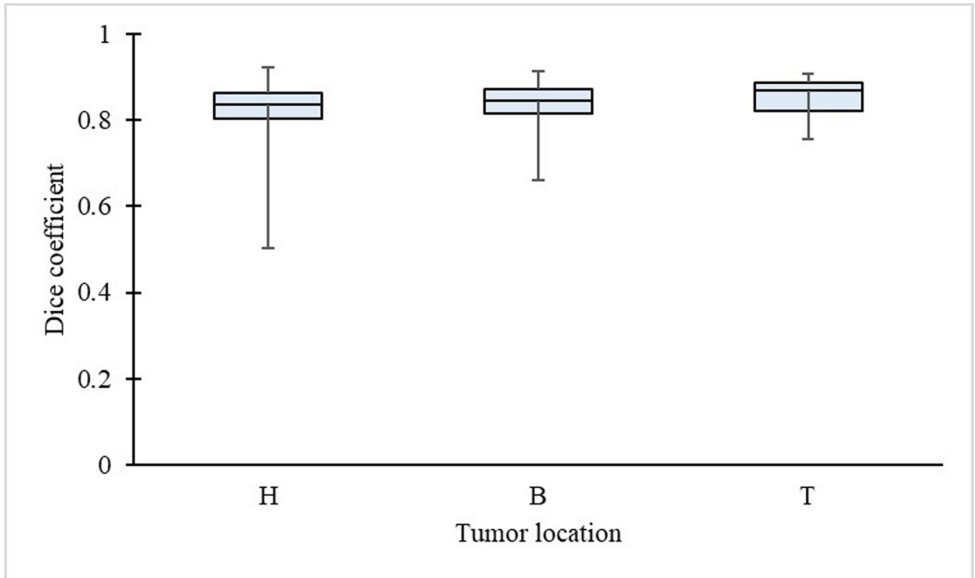

**Fig 5. Tumor location and Dice coefficient of the pancreas.** H means head, B means body, T means tail.

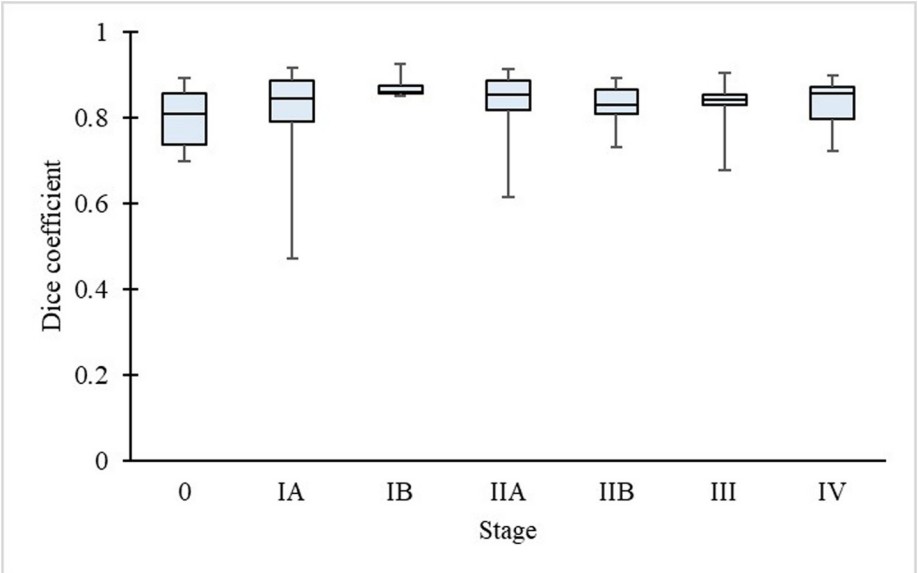

**Fig 6. Stage of disease and Dice coefficient of the pancreas.**

boundaries between the tumor and pancreatic parenchyma could be cited as causes of deterioration in extraction.

With regard to the sensitivity in extracting each vessel around the pancreas, deterioration in extraction accuracy was observed in the posterior superior pancreatic duodenal vein, left gastric vein, and posterior inferior pancreatic duodenal artery. This may be because these blood vessels often deviate in their courses, and they display much movement near the tumor,

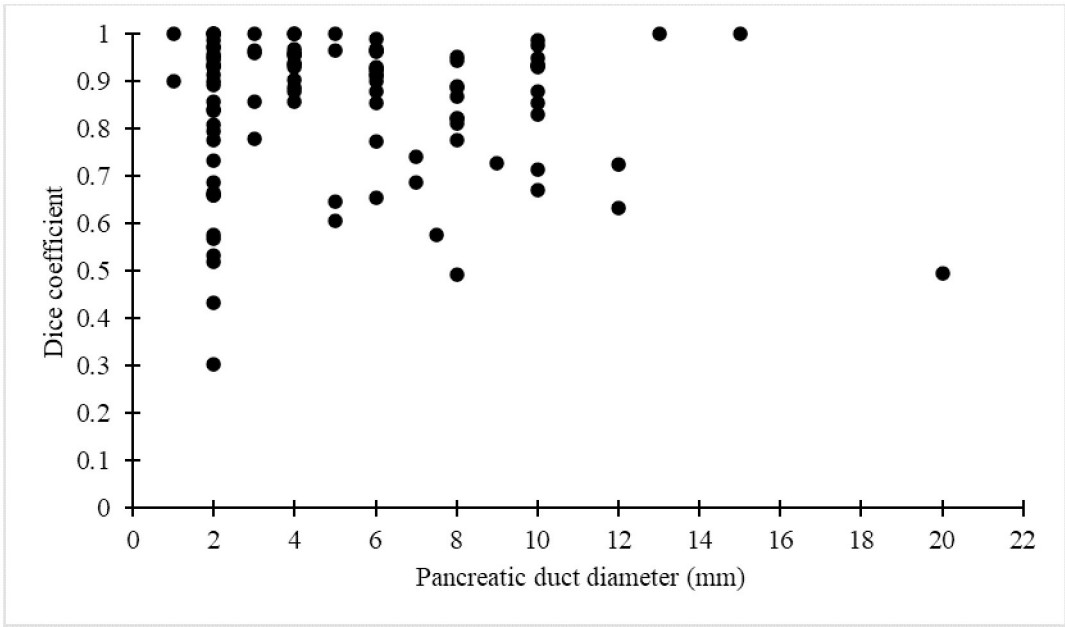

**Fig 7. Correlation between pancreatic duct diameter and Dice coefficient of the pancreatic duct.**

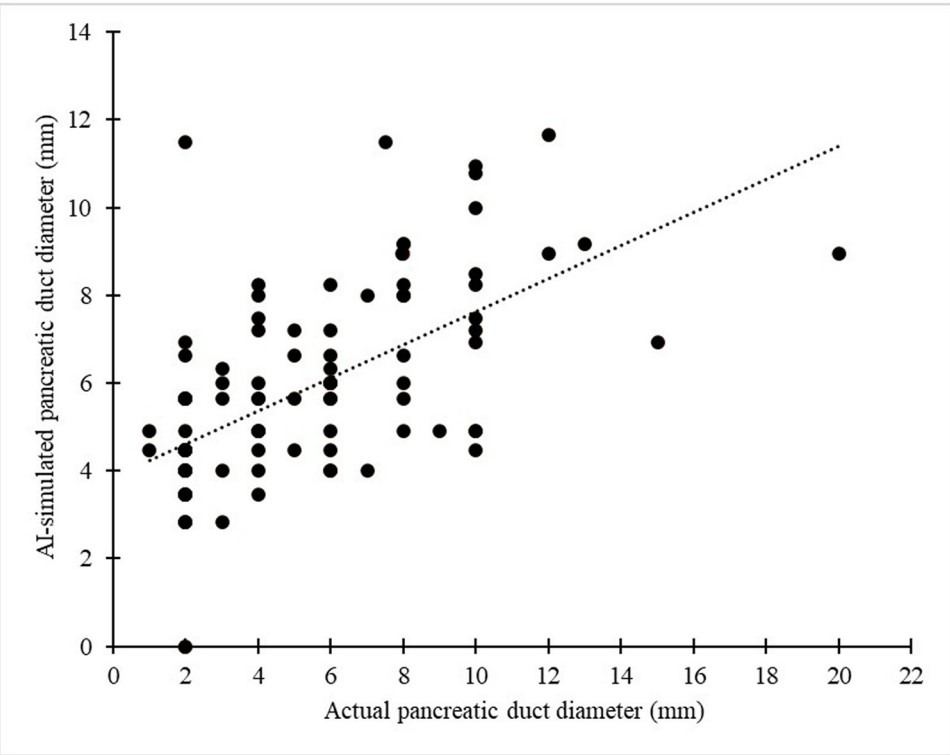

**Fig 8. Correlation between AI-simulated diameter and actual diameter of the pancreatic duct.**

which makes them susceptible to tumor effects. Due to advancements in image analysis in recent years, the importance of preoperatively understanding how the posterior inferior pancreatic duodenal artery and jejunal vein move in pancreatic resection has been reported [16–18]. Continued accumulation of AI learning data is needed to improve the sensitivity of vascular extraction, even if tumor effects are present.

Even in pancreatic cancer patients, no changes were observed in the extraction accuracy of the AI engine for stage, size, or tumor position. Although confirmation by a surgeon is needed, there is no associated labor cost, as was required with previous software, so image extraction can be performed semiautomatically and can be widely used in pancreatic resection. Furthermore, pancreatic resection may be possible using 3D images that visualize the peripancreatic anatomy extracted by the AI engine. The pancreas can be isolated in an arbitrary position, and the anatomical relationship of the isolated pancreas with the surrounding vessels and surrounding organs, predicted pancreatic duct position in relation to the isolated pancreas, automatic calculations of the excised pancreatic and residual pancreatic regions, shape of the surface of the resected pancreas, surface area, and thickness of the excised pancreas can also be obtained automatically from the 3D image.

In particular, with a preoperative understanding of pancreatic duct dissection parameters, including the pancreatic duct diameter, the pancreas can be dissected along the left edge of the portal vein in many cases during standard pancreatectomy procedures, such as PD and distal pancreatectomy, and the dissection line can be easily simulated. Indeed, we previously mentioned the importance of preoperative simulations of pancreatic duct dissection [13]. However, it is difficult to know the three-dimensional position and thickness of a pancreatic duct based on the position of an isolated pancreas before the procedure, so these data must be obtained

during the procedure. However, in cases of a normal pancreas without pancreatic duct dilation, it is difficult to identify the position of the pancreatic duct during the procedure. Therefore, searching for the position of the pancreatic duct may lead to damage to the pancreatic duct or prolongation of the procedure duration. Understanding the pancreatic duct resection approach before surgery with the help of 3D images from the AI engine can help prevent such situations and may provide useful information for the surgical team [13, 17]. In addition, 3D images from the AI engine make it possible to share anatomical images from any point of view and are said to be highly useful in laparoscopic pancreatectomy and robot-assisted pancreatectomy. Moreover, as our previous research has already shown, sharing of the anatomical images with the surgical staff contributed to the reduced intraoperative blood loss in conjunction with the mastery of surgical techniques, advances in surgical instruments and perioperative management in pancreatic surgery [13, 17, 19].

Another clinical benefit of creating preoperative 3D images was assumed to evaluate the residual pancreatic volume. Our previous studies have already reported that postoperative pancreatic volume was closely associated with postoperative pancreatic endocrine insufficiency or useful value for predicting postoperative long-term outcomes in pancreatic cancer patients [11, 12]. Therefore, the early identification of these high-risk patients can increase the accuracy of the perioperative benefit/risk assessment and the information given to patients. We suggested that these patients should be more carefully monitored for the presence of postoperative pancreatic endocrine insufficiency or the recurrence of pancreatic cancer during the postoperative follow-up period.

We conclude that extraction of the pancreatic parenchyma, surrounding vessels, and surrounding organs with the SYNAPSE VINCENT® AI engine can be useful as surgical support. In the future, we plan to apply this method to surgical navigation with an expanded AI engine and to diagnostic imaging technology using AI to identify early pancreatic cancer lesions.

## Supporting information

**S1 Data.**
(XLSX)

**S2 Data.**
(XLSX)

## Acknowledgments

We thank the Department of Radiology, Saitama Cancer Center, for consulting on the radiological findings.

## Author Contributions

**Conceptualization:** Ryoichi Miyamoto.

**Data curation:** Ryoichi Miyamoto, Kei Kitamura, Hiroyuki Ishida, Shinichi Matsudaira.

**Formal analysis:** Ryoichi Miyamoto, Aya Ogasawara.

**Funding acquisition:** Ryoichi Miyamoto.

**Investigation:** Ryoichi Miyamoto, Aya Ogasawara, Kei Kitamura, Hiroyuki Ishida.

**Methodology:** Ryoichi Miyamoto.

**Project administration:** Ryoichi Miyamoto.

**Resources:** Ryoichi Miyamoto.

**Software:** Ryoichi Miyamoto.

**Supervision:** Ryoichi Miyamoto, Amane Takahashi, Toshiro Ogura, Satoshi Nozu, Yoshiyuki Kawashima.

**Validation:** Ryoichi Miyamoto.

**Visualization:** Ryoichi Miyamoto.

**Writing – original draft:** Ryoichi Miyamoto.

**Writing – review & editing:** Ryoichi Miyamoto.

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
