## [Decision Letter · Decision Letter 0]

26 Sep 2022

PONE-D-22-19378Three-dimensional simulation of the pancreatic parenchyma, pancreatic duct and vascular arrangement in pancreatic surgery using a deep learning algorithmPLOS ONE

Dear Dr. Miyamoto,

Thank you for submitting your manuscript to PLOS ONE. After careful consideration, we feel that it has merit but does not fully meet PLOS ONE’s publication criteria as it currently stands. Therefore, we invite you to submit a revised version of the manuscript that addresses the points raised during the review process.

We look forward to receiving your revised manuscript.

Kind regards,

Ulrich Wellner, Prof Dr. med.

Academic Editor

PLOS ONE

Journal Requirements:

“This research is a cooperative study with Fujifilm Medical Co., Ltd., Tokyo, Japan.

MR received a specific grant from Fujifilm Medical Co., Ltd., Tokyo, Japan.

MR and Fujifilm Medical Co., Ltd., Tokyo, Japan had data analysis together.”

“I have read the journal's policy and the authors of this manuscript have the following competing interests: This research is a cooperative study with Fujifilm Medical Co., Ltd., Tokyo, Japan.”

Reviewers' comments:

Reviewer's Responses to Questions

**Comments to the Author**

1. Is the manuscript technically sound, and do the data support the conclusions?

Reviewer #1: Yes

Reviewer #2: Partly

2. Has the statistical analysis been performed appropriately and rigorously? 

Reviewer #1: Yes

Reviewer #2: Yes

3. Have the authors made all data underlying the findings in their manuscript fully available?

Reviewer #1: Yes

Reviewer #2: Yes

4. Is the manuscript presented in an intelligible fashion and written in standard English?

Reviewer #1: Yes

Reviewer #2: Yes

5. Review Comments to the Author

Reviewer #1: This study evaluates the performance of SYNAPSE VINCENT in the semiautomated surgical simulation of the pancreatic parenchyma, pancreatic ducts, and peripancreatic vessels using AI with deep learning algorithms. Overall, this is an interesting study.

The study found that the visualization of the pancreatic duct in the control group was significantly lower than that in the pancreatic cancer group. The simulation success rate of the posterior inferior pancreatic duodenal artery and posterior superior pancreatic duodenal vein was significantly lower than that of the control group, which was in line with clinical expectations. These data are consistent with our clinical experience.

The potential benefit of this study is that reconstruction of the pancreatic duct can provide surgeons with information on the location of the pancreatic duct, which is helpful for finding the main pancreatic duct during surgery. Even for non-tumor pancreatic tissue, the main pancreatic duct reconstruction rate of this model can still reach 0.77 (the main pancreatic duct of normal pancreas is often difficult to find during surgery), which may be helpful for pancreaticojejunostomy.

The work of this paper is clear and logical. I think the paper is publishable. Howerer，there are still some questions should be addressed before publication:

1.This study focused on the reconstruction of the main pancreatic duct and the blood vessels around the pancreas, but did not discuss the value of this model for evaluating the relationship between blood vessels and tumors, such as whether the tumor invades blood vessels, and how effective is its evaluation? Does this model have advantages over CT? This may be a question of greater interest to surgeons.

2. In the Discussion section, the authors should focus more on discussing the clinical benefits and potential applications of the model rather than simply presenting reconstructed data.

Reviewer #2: The study evaluates the performance of SYNAPSE VINCENT ® in the semiautomated surgical simulation of the pancreatic parenchyma, pancreatic ducts, and peripancreatic vessels using an artificial intelligence (AI) engine designed with deep learning algorithms.

And the result show that Extraction of the pancreatic parenchyma, pancreatic duct, and surrounding vessels with the SYNAPSE VINCENT ® AI engine assumed to be useful as surgical simulation.

Manuscript was well prepared. The topic of this manuscript is meaningful, which focus on the construct of 3D images of vasculature and organs.

some sentences should be revised，such as “the Dice coefficients of pancreatic parenchyma and pancreatic duct extraction were 0.83/0.86 and 0.84/0.77". P values should be added.

6. PLOS authors have the option to publish the peer review history of their article (what does this mean?). If published, this will include your full peer review and any attached files.

Reviewer #1: No

Reviewer #2: No

---

## [Author Response · Author response to Decision Letter 0]

30 Sep 2022

RESPONSES TO THE REVIEWERS

October 1, 2022

Dear Editor,

Title: Three-dimensional simulation of the pancreatic parenchyma, pancreatic duct and vascular arrangement in pancreatic surgery using a deep learning algorithm

Authors: Ryoichi Miyamoto, Amane Takahashi, Aya Ogasawara, Toshiro Ogura, Kei Kitamura, Hiroyuki Ishida, Shinichi Matsudaira, Satoshi Nozu, Yoshiyuki Kawashima

Name of Journal: PLOS ONE

Manuscript NO: PONE-D-22-19378

The manuscript has been revised according to the reviewers’ suggestions. The modified parts are highlighted in red font. Please see the highlighted sentences in the annotated manuscript file. Thank you.

Reviewer Comments:

Reviewer #1:

1.This study focused on the reconstruction of the main pancreatic duct and the blood vessels around the pancreas but did not discuss the value of this model for evaluating the relationship between blood vessels and tumors, such as whether the tumor invades blood vessels, and how effective is its evaluation? Does this model have advantages over CT? This may be a question of greater interest to surgeons.

We thank the reviewer for this comment. In accordance with the reviewer’s comment, it is also important to evaluate the relationship between blood vessels and tumors in pancreatic surgery. Pancreatic tumors, including pancreatic cancer, have an unclear contrast enhancement effect using dynamic MDCT images. Semiautomatic image extraction of pancreatic tumors is very difficult. However, by performing anatomical simulation using a 3D image including a manually masked tumor area, it is possible to grasp the positional relationship between the tumor position and the blood vessel in a three-dimensional view and to plan the surgical method, including consideration of the blood vessel to be resected together.

Furthermore, the surgical team was able to view the preoperative 3D image on a large display during the actual surgery, enabling discussion of the critical points of the surgical procedure. We assumed that these findings have a great advantage over MDCT.

To achieve the function of automatically extracting the tumor area, continued accumulation of AI learning data is needed.

2. In the Discussion section, the authors should focus more on discussing the clinical benefits and potential applications of the model rather than simply presenting reconstructed data.

We thank the reviewer for this comment. In accordance with the reviewer’s comment, we revised the manuscript on Page 14, Lines 23 to Page 15, Lines 12.

In terms of the clinical benefits and potential applications, we assumed that sharing of the anatomical images with the surgical staff contributed to the reduced intraoperative blood loss in conjunction with the mastery of surgical techniques, advances in surgical instruments and perioperative management in pancreatic surgery.

Another clinical benefit of creating preoperative 3D images was assumed to be evaluating the residual pancreatic volume. Our previous studies have already reported that postoperative pancreatic volume was closely associated with postoperative pancreatic endocrine insufficiency or useful value for predicting postoperative long-term outcomes in pancreatic cancer patients. Therefore, the early identification of these high-risk patients can increase the accuracy of the perioperative benefit/risk assessment and the information given to patients. We suggest that these patients should be more carefully monitored for the presence of postoperative pancreatic endocrine insufficiency or the recurrence of pancreatic cancer during the postoperative follow-up period.

Reviewer #2:

“the Dice coefficients of pancreatic parenchyma and pancreatic duct extraction were 0.83/0.86 and 0.84/0.77". P values should be added.

We thank the reviewer for this comment. In accordance with the reviewer’s comment, we revised the manuscript on Page 10, Lines 8, 22, 23 and Page 11, Lines 5.

We sincerely appreciate the reviewers’ comments, which have helped us improve the paper and provided new insights for our study. We hope that the revised manuscript is now suitable for publication in PLOS ONE.

Ryoichi Miyamoto

Department of Gastroenterological Surgery, Saitama Cancer Center, 780 Komuro, Ina-machi, Kita-Adachi-gun, Saitama 362-0806, Japan

Phone: +81-48-722-1111; Fax: +81-48-722-1129

Email: goodfirst883@gmail.com

---

## [Editor Report · Decision Letter 1]

11 Oct 2022

Three-dimensional simulation of the pancreatic parenchyma, pancreatic duct and vascular arrangement in pancreatic surgery using a deep learning algorithm

PONE-D-22-19378R1

Dear Dr. Miyamoto,

We’re pleased to inform you that your manuscript has been judged scientifically suitable for publication and will be formally accepted for publication once it meets all outstanding technical requirements.

Kind regards,

Ulrich Wellner, Prof Dr. med.

Academic Editor

PLOS ONE

Additional Editor Comments (optional):

Thank you. The reviewers' comments have been sufficiently addressed.
---

## [Editor Report · Acceptance letter]

19 Oct 2022

PONE-D-22-19378R1 

Three-dimensional simulation of the pancreatic parenchyma, pancreatic duct and vascular arrangement in pancreatic surgery using a deep learning algorithm 

Dear Dr. Miyamoto:

I'm pleased to inform you that your manuscript has been deemed suitable for publication in PLOS ONE. Congratulations! Your manuscript is now with our production department. 

Kind regards, 

on behalf of

Mr. Ulrich Wellner 

Academic Editor

PLOS ONE